# Early warning signals do not predict a warming-induced experimental epidemic

**Madeline Jarvis-Cross** [1]*, **Devin Kirk** [2,3], **Leila Krichel** [1], **Pepijn Luijckx** [4],
**Péter K. Molnár** [1,5], **Martin Krkošek** [1]

**1** Department of Ecology and Evolutionary Biology, University of Toronto, Toronto, Ontario, Canada,
**2** Department of Biology, Stanford University, Stanford, California, United States of America, **3** Department of Zoology and Biodiversity Research Centre, University of British Columbia, Vancouver, British Columbia, Canada, **4** School of Natural Sciences, Trinity College Dublin, Dublin, Ireland, **5** Department of Biological Sciences, University of Toronto Scarborough, Scarborough, Ontario, Canada

\* madeline.jarvis.cross@mail.utoronto.ca

## Abstract

Climate change can impact the rates at which parasites are transmitted between hosts, ultimately determining if and when an epidemic will emerge. As such, our ability to predict climate-mediated epidemic emergence will become increasingly important in our efforts to prepare for and mitigate the effects of disease outbreaks on ecological systems and global public health. Theory suggests that statistical signatures termed "early warning signals" (EWS), can function as predictors of epidemic emergence. While a number of works report post hoc detections of EWS of epidemic emergence, the theory has yet to be experimentally tested. Here, we analyse experimental and simulated time series of disease spread within populations of the model disease system *Daphnia magna–Ordospora colligata* for EWS of climate-mediated epidemic emergence. In this system, low temperatures prevent disease emergence, while sufficiently high temperatures force the system through a critical transition to an epidemic state. We found that EWS of epidemic emergence were nearly as likely to be detected in populations maintained at a sub-epidemic temperature as they were to be detected in populations subjected to a warming treatment that induced epidemic spread. Our findings suggest that the detection of false positives may limit the reliability and utility of EWS as predictors of climate-mediated epidemic emergence.

## Introduction

Host-parasite dynamics are often mediated by temperature [1–3]. For example, parasite replication, development, and mortality rates [4–6], parasite infectivity and virulence [7], host resistance [8], immune functioning [9–11], and tolerance [12], and host-parasite contact rates [13,14] are all sensitive to temperature. As such, climate change can impact the rates at which parasites are transmitted between hosts, and ultimately, affect if and when epidemics emerge [1,15–17]. For example, climate

**Data availability statement:** Empirical and simulated time series data, and scripts available from GitHub at https://github.com/MadelineJC/EarlyWarningSignals_Repo.

**Funding:** This work was supported by The Emerging and Pandemic Infections Consortium Doctoral Award (MJ-C), the Ontario Graduate Scholarship (MJ-C, DK), Natural Sciences and Engineering Research Council of Canada Discovery (RGPIN-2024-05054) and Accelerate Grants (MK), a Canada Research Chair (MK), and an Institutional Strategic Fund from the Wellcome Trust to Trinity College Dublin (PL). PKM is grateful for support from a Natural Sciences and Engineering Research Council of Canada (NSERC) Discovery Grant (RGPIN-2023-05331), the Canada Foundation for Innovation (CFI) John R. Evans Leaders Fund (Grant Number 35341) and the Ministry of Research, Innovation and Sciences (MRIS) Ontario Research Fund. The funders had no role in study design, data collection and analysis, decision to publish, or preparation of the manuscript.

**Competing interests:** The authors have declared that no competing interests exist.

change can alter species ranges and population densities, thereby altering contact rates, and thus, parasite transmission rates [18,19]. In vector-borne disease systems like malaria and dengue fever, climate change may increase vector range and competence, though subsequent impacts on incidence and epidemic emergence have been debated [20–22]. Climate change can also asymmetrically affect vital rates within hosts and their parasites, widening performance gaps between them, and changing infection patterns [23,24]. In amphibians and a variety of marine organisms (e.g., shellfish, corals, and finfish), climate change has been linked to decreased host immune function and/or increased pathogen virulence, resulting in widespread disease-mediated declines [11,25,26]. In contrast, increasing mean temperatures have also been associated with decreased infectivity of cold-water pathogens in salmonids [27]. In general, a wide array of studies of terrestrial and aquatic systems have shown system-specific impacts of climate change on disease spread and emergence [1,2,28]. As Earth's climate changes, our ability to predict climate-mediated epidemic emergence will become increasingly important in our efforts to prepare for and mitigate the effects of disease outbreaks on ecological systems, agriculture, and human health. It has been proposed that such predictions may be made by leveraging the statistical signatures associated with ecological and epidemiological time series data.

As dynamical systems, ecological systems are often characterized by critical thresholds [29,30]. When parameters that define a system change, the system may pass through a local bifurcation and experience a critical transition, whereby the stability of an equilibrium changes, or the system quickly shifts among alternative stable states. As a system approaches a local bifurcation, its resilience decreases, and the rate at which it recovers from small perturbations slows down (termed "critical slowing down") [31–33]. Theory suggests that as a system's resilience decreases, it will produce statistically anomalous behaviors, such as increasing variation and temporal autocorrelation. These statistical signatures have been termed "early warning signals" (EWS), and, when detected, have been employed as a means of predicting upcoming critical transitions [29,31].

Although contentious [34,35], the detection of EWS has been used to assert the predictability of critical transitions in a broad range of dynamical systems, including climate systems [36–39], freshwater systems [40–43], and infectious disease systems [44–49]. In epidemiology, when the basic reproductive number ($R_0$) exceeds a value of one, infectious disease systems pass through a transcritical bifurcation, and shift from a sub-epidemic state to an epidemic state [48,50]. As such, these methods have been employed to analyse prevalence time series of past epidemics for EWS, and determine if epidemic (re-)emergence could have been predicted [51]. For example: Harris et al. (2020) analysed long-term malaria incidence data and detected EWS in periods preceding epidemic emergence in Kericho, Kenya between 1981 and 1993. While studies of different disease systems have yielded varying results [51–53], the absence of "control" populations within which sub-epidemic spread fails to transition into epidemic spread limits our ability to interpret these results, and raises questions about the reliability and utility of EWS as tools for predicting and mitigating the impacts of epidemic emergence events. It has been additionally posited

that the study of observed systems known to exhibit critical transitions introduces a kind of selection bias, whereby the likelihood of observing evidence of a phenomenon is confused for the likelihood of the phenomenon, given the evidence [34].

Building off of Kirk et al. (2020), we analysed experimental and simulated time series of disease spread within populations of the model disease system *Daphnia magna–Ordospora colligata* for EWS of critical transition to an epidemic state. During Kirk et al.'s (2020) experiment, control populations were maintained at a sub-epidemic temperature and test populations were subjected to a warming treatment that forced the system through a critical transition to an epidemic state. We found that EWS of epidemic emergence were nearly as likely to be detected in control populations as they were to be detected in warming populations, implying that the ubiquity of false positive detections may limit the reliability of EWS as predictors of climate-mediated epidemic emergence.

## Methods

First, we will describe Kirk et al.'s (2020) experiment and the resulting population level infection data. Then, we will describe Kirk et al.'s (2020) trait-based mechanistic model of *O. colligata* transmission dynamics, and how we used this model to simulate time series of disease spread under constant and warming conditions. Finally, we will describe how we analysed the experimental and simulated data for early warning signals of disease emergence, and four ways in which we evaluated the reliability of our empirical detections.

### Warming experiment

The empirical data were derived from a 120-day experiment involving eight *Daphnia magna* populations and their environmentally transmitted microsporidian parasite, *Ordospora colligata* [14]. *D. magna* is a freshwater planktonic crustacean often used as a model organism to develop and evaluate ecological theories [54]. Since the early 1990s, *D. magna* has been used to study host–parasite dynamics at the within- and between-host levels [54]. In nature, *D. magna* hosts numerous parasites, including a microsporidian parasite, *O. colligata* [55]. *D. magna* become infected when they passively ingest the parasite while grazing. The parasite infects gut epithelial cells and replicates into clusters of spores. After some period of time, infected epithelial cells burst, releasing spores to infect new cells, or into the water column where they can be ingested by a new host.

Prior to the beginning of the experiment, uninfected *D. magna* were maintained at 20°C for fifteen days, and subsequently, 10°C for an additional fifteen days to acclimatize the populations to the starting conditions of the experiment. Population abundances stabilized between 150 and 240 large individuals per population. The experiment was initiated by introducing three randomly selected adult *D. magna* from infected stock populations to each of the eight experimental populations. Disease prevalence in infected stock populations was approximately 46.5%. Every third day of the experiment, twelve large females were removed from each experimental population and assessed for infection. Sampled individuals were then replaced by three randomly selected individuals from infected stock populations, and nine individuals from uninfected stock populations to maintain an immigration rate of infected individuals and to maintain population abundance. After sampling and replacement, (1) three liters of *D. magna* growth medium at 5% of the recommended SeO$_2$ (ADaM [56]) were removed from each experimental population and replaced with three liters of new ADaM to refresh the growth medium, resulting in a *per capita* spore mortality rate of *O. colligata* of $\gamma = 0.0286$ days$^{-1}$ (assuming that the spores were well mixed) [14], and (2) each population was fed 350 million batch cultured algae (*Monoraphidium minutum*).

Populations 1–4 were designated as controls and maintained at a constant temperature of 10°C, which is too cold for the parasite to establish and spread (Figs 1 and 2). Populations 5–8 were assigned to a warming treatment, wherein the temperature of the system was set to 10°C at the beginning of the experiment, and increased by 0.5°C every fifteen days, to a maximum of 13.5°C after 105 days. All experiments were terminated after 120 days (Figs 1 and 2). Kirk et al.'s (2020)

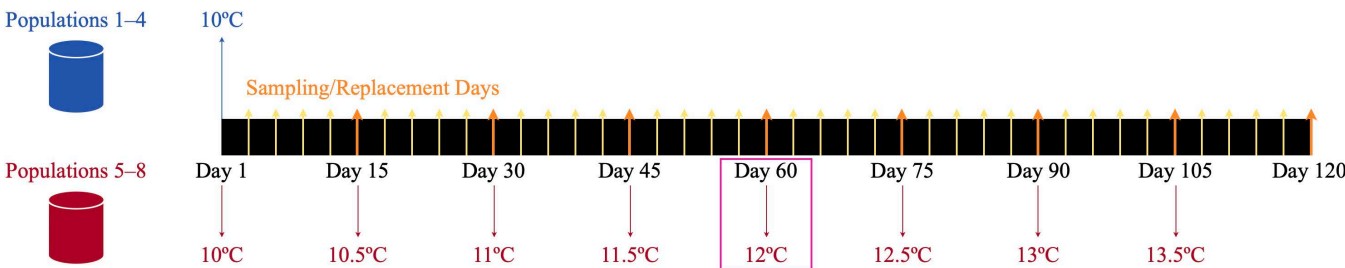

**Fig 1. Schematic detailing experimental design.** Populations 1–4 were held at 10°C for the duration of the experiment (shown in blue). Populations 5–8 were subjected to a gradual warming treatment (shown in red). Every third day of the experiment, twelve individuals were sampled from each population, and replaced by three randomly selected individuals from infected stock populations, and nine individuals from uninfected stock populations. Sampled individuals were assessed for infection. The pink box represents the day on which the temperature of the system was increased to 12°C, allowing $R_0$ to surpass the critical threshold, pushing warming populations from a sub-epidemic state to an epidemic state.

trait-based mechanistic model predicts that $R_0$ will exceed the critical threshold of $R_0 = 1$ at 12°C, pushing the system through a critical transition from a sub-epidemic state to an epidemic state. As such, only warming populations were subjected to epidemic emergence. This prediction is supported by the experimental data [14].

## Model description

The mechanistic model (Eqns. (1.1)–(1.4)) contains thirteen parameters, six of which are temperature-dependent ($T$), and describes how susceptible hosts ($S$) transition into infecteds ($I$), and subsequently, dead infecteds ($D$) (Table 1). Infecteds ($I$) and dead infecteds ($D$) shed parasite spores into the environment ($E$) (Table 1). All parameters and temperature-dependencies were set as determined by Kirk et al., 2018 and Kirk et al., 2019 [6,13,14]. For a detailed description of model parameterization and model assumptions, please refer to Kirk et al.'s (2020) Supplementary Material. Per the mechanistic model, susceptible hosts ($S$) were added to each population by the experimenter and as described above ($\phi_s$) or through density-dependent recruitment (wherein $\psi$ represents maximum recruitment rate and $K$ represents carrying capacity), and lost via infection (at a rate of $\chi(T)\sigma(T)SE$), natural mortality ($\mu$), or harvesting ($h$). Infected adults ($I$) were added via immigration ($\phi_I$) and transmission ($\chi(T)\sigma(T)SE$), and lost via natural and parasite-induced mortality ($\mu(T)$; $\alpha(T)$), and harvesting ($h$). Dead infecteds ($D$) were added via natural and parasite-induced mortality ($\mu(T)$; $\alpha(T)$), and were lost via degradation ($\theta$). Spores were released into the environment ($E$) via shedding from living and deceased infecteds ($\lambda(T)$; $\omega(T)$) and lost via experimentally induced mortality ($\gamma$).

$$\frac{dS}{dt} = \phi_s + \psi(S+I)\left(1 - \left(\frac{S+I}{K}\right)\right) - \chi(T)\sigma(T)SE - \mu(T)S - hS \tag{1.1}$$

$$\frac{dI}{dt} = \phi_I + \chi(T)\sigma(T)SE - (\mu(T) + \alpha(T))I - hI \tag{1.2}$$

$$\frac{dD}{dt} = (\mu(T) + \alpha(T))I - \theta D \tag{1.3}$$

$$\frac{dE}{dt} = \lambda(T)I + \omega(T)\theta D - \gamma E \tag{1.4}$$

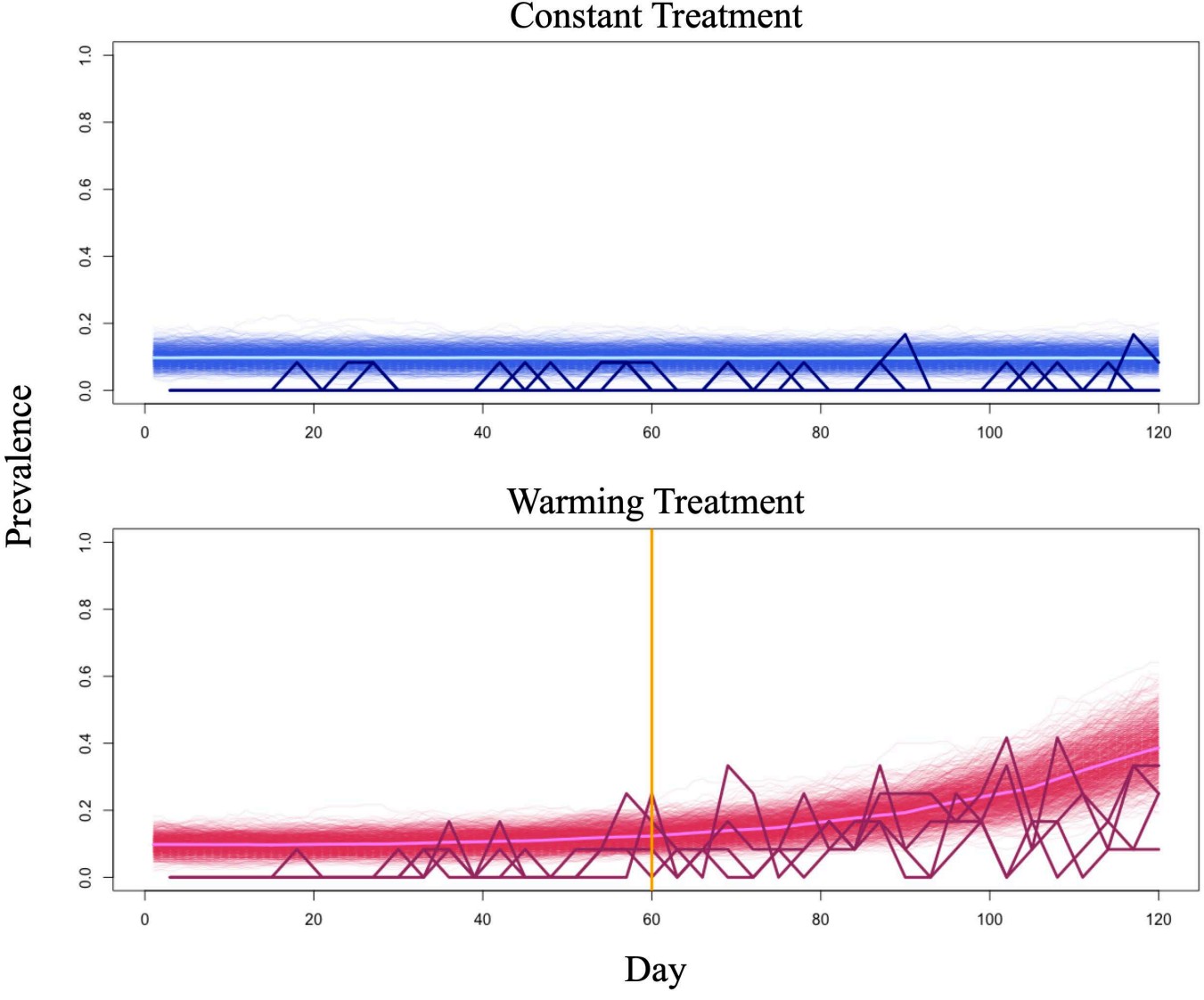

**Fig 2. Empirical and simulated time series.** In both plots, simulated time series are shown in a lighter shade, while experimental time series are shown in a darker shade. Mean simulated prevalence is shown in the lightest shade. The orange vertical line denotes the day on which the temperature of the system was increased to 12°C, allowing $R_0$ to surpass the critical threshold, pushing warming populations from a sub-epidemic state to an epidemic state.

## Simulation experiment

To generate the simulated data, we conducted demographically stochastic simulations of the mechanistic model that matched the duration and environmental conditions of the experimental system. Under constant conditions, the temperature of the system was set to 10°C for the duration of the simulation. Under warming conditions, the temperature of the system was set to 10°C, and increased by 0.5°C every fifteen days. A maximum temperature of 13.5°C was reached after 105 days, and the experiment was terminated after 120 days (Fig 1). We generated one thousand time series under

**Table 1. Parameter definitions and values for the mechanistic *D. magna–O. colligata* model (Eqns. (1.1)–(1.4)), replicated from Kirk et al. (2020). Note that the last six parameters are temperature-dependent and are written as functions of temperature, *T*.**

| Parameter | Description | Source | Function | Value |
|---|---|---|---|---|
| $\phi_S$ | Total rate of input of susceptibles | Kirk et al. (2020) | Constant | 3.535 day$^{-1}$ |
| $\phi_I$ | Total rate of input of infecteds | Kirk et al. (2020) | Constant | 0.465 day$^{-1}$ |
| $K$ | Adult density-dependent recruitment constraint | Kirk et al. (2020) | Constant | 170 |
| $\psi$ | Maximum recruitment | Kirk et al. (2020) | Constant | 1.33 day$^{-1}$ |
| $h$ | Harvesting | Kirk et al. (2020) | Constant | 0.0235 day$^{-1}$ |
| $\gamma$ | Environmental spore mortality | Kirk et al. (2020) | Constant | 0.0286 day$^{-1}$ |
| $\theta$ | Corpse degradation | Kirk et al. (2020) | Constant | 0.1 day$^{-1}$ |
| $\mu(T)$ | Natural mortality rate | Kirk et al. (2018) | Sharpe-Schoolfield[a] | day$^{-1}$ |
| $\chi(T)$ | Contact rate | Kirk et al. (2019) | Sharpe-Schoolfield[a] | day$^{-1}$ |
| $\sigma(T)$ | Probability of infection | Kirk et al. (2019) | Sharpe-Schoolfield[a] | – |
| $\lambda(T)$ | Parasite shedding rate | Kirk et al. (2018) | Sharpe-Schoolfield[a] | day$^{-1}$ |
| $\alpha(T)$ | Parasite-induced mortality rate | Kirk et al. (2018) | Sharpe-Schoolfield[a] | day$^{-1}$ |
| $\omega(T)$ | Parasite intensity at host death | Kirk et al. (2018) | Sharpe-Schoolfield[a] | – |

[a] See Fig 1 in Kirk et al., (2020) for functional forms.

constant conditions, and one thousand time series under warming conditions (Fig 2). Demographic stochasticity was introduced via the Gillespie algorithm using the 'GillespieSSA' package in R version 4.2.2 [57]. The GillespieSSA package uses the model of interest (here, Eqns. (1.1)–(1.4)) to generate stochastic population trajectories. Neither the simulations nor the experiment included environmental stochasticity.

## Analysis

**Statistical metrics.** We considered ten EWS of the transcritical bifurcation at $R_0 = 1$ (Table 2). In addition to commonly studied statistics (e.g., variance and autocorrelation, which are expected to increase as a system approaches a local bifurcation), we considered mean prevalence, which we expected to increase as $R_0$ increased towards one [45,58]. We also considered: the coefficient of variation, which relates mean and standard deviation; the index of dispersion, which relates mean and variance; first difference variance; autocovariance, which is the covariance of the process with itself and is related to variance and autocorrelation; and decay time, a log-transform of autocorrelation (Table 2) [45,58]. We also considered two higher-order moments, skewness and kurtosis. Skewness is a measure of the symmetry of the frequency-distribution curve, and kurtosis is a measure of the sharpness of the peak of the frequency-distribution curve (Table 2).

**Pre-processing time series data.** While it is common practice to pre-process time series data to remove seasonal and longer-term trends [51,59], neither our experimental nor our simulated data were influenced by seasonal forcing, and were collected/generated over relatively short time periods. As such, "detrending" these data may introduce artificial patterns rather than remove them [46,60]. In the interest of robustness, we ran analyses on raw and pre-processed data (with pre-processing done using a Gaussian kernel [37,45,48,49,51,61]). We found that pre-processing inflated the strength and reversed the directionality of observed trends, suggesting that pre-processing introduced artificial patterns [46]. (See S1 Text, S1–S4 Figs for a more detailed description of these methods and their results.) As such, we will only present and discuss the results of analyses on raw data.

**Sliding windows and Kendall's Tau.** We used the 'evobiR' package [62] in R version 4.2.2 to calculate mean, standard deviation, skewness, and kurtosis. We used the 'stats' package in R version 4.2.2 to calculate autocorrelation and autocovariance, and calculated the coefficient of variation, index of dispersion, and first difference variance according to their mathematical definitions, as stated in Table 2 [58]. All statistical metrics were calculated within sliding windows,

**Table 2. Statistical metrics used to evaluate time series for critical slowing down.**

| Statistic | Mathematical Definition[a] | Expected Trend in Metric as Time Series Approaches Transcritical Bifurcation[a] |
|---|---|---|
| Mean | $\mu_t = E[X_t]$ | Increase |
| Skewness | $Skew_t = \frac{E[(X_t - \mu_t)^3]}{\sigma_t^3}$ | Increase |
| Kurtosis | $Kurt_t = \frac{E[(X_t - \mu_t)^4]}{\sigma_t^4}$ | Increase |
| Variance | $\sigma_t^2 = E[(X_t - \mu_t)^2]$ | Increase |
| Coefficient of Variation | $CoV_t = \frac{\sigma_t}{\mu_t}$ | No Change |
| Index of Dispersion | $IoD_t = \frac{\sigma_t^2}{\mu_t}$ | Increase |
| First Difference Variance | $\Delta \sigma_t^2 = \sigma_t^2 - \sigma_{t-1}^2$ | Increase |
| Autocorrelation | $AC_t(1) = \frac{ACoV_t(1)}{\sigma_t \sigma_{t-1}}$ | Increase |
| Autocovariance | $ACoV_t(1) = E[(X_t - \mu_t)(X_{t-1} - \mu_{t-1})]$ | Increase |
| Decay Time | $\overline{\tau}_t = \frac{-1}{ln[AC_t(1)]}$ | Increase |

[a] As stated in Brett et al., 2018 $E[X_t]$ refers to the expected value of $X$ at time $t$.

leading up to day sixty (as we expect rising temperatures to induce a bifurcation on day sixty in warming populations). We analysed the empirical data within fifteen-day sliding windows, as temperatures were increased every fifteen days in warming populations, and analysed the simulated data within five-, fifteen-, and thirty-day sliding windows to assess the sensitivity of outcomes to sliding window choice [37]. Given that experimental sampling occurred every three days, we were unable to apply a five-day sliding window. Relatedly, applying a thirty-day sliding window to the first half of the experimental time series yielded just ten observations. Despite this, in the interest of robustness, we applied six- and thirty-day windows. While the six-day results bore some resemblance to the fifteen-day results, the thirty-day results often exhibited an unexpected negative association between statistical metrics and time, suggesting they may be unreliable (S5 Fig). To evaluate trends in statistics during the approach to criticality, we calculated Kendall's rank correlation coefficient (or, the trend coefficient). Given that the temperature of the system increased to 12°C on day sixty, we considered three pre-critical intervals: days one to sixty (sixty days), days twenty to sixty (forty days), and days thirty to sixty (thirty days). While data quality precluded the reliable analysis of empirical time series within the forty- and thirty-day pre-critical intervals, analyses of simulated time series were not substantially affected by pre-critical interval choice. As such, we present analyses of empirical time series performed within the sixty-day pre-critical interval in the main text. Kendall's $\tau$ ($\{\tau \in R \mid -1 \leq \tau \leq 1\}$) is a measure of the ordinal association between two objects [63,64]. Here, Kendall's $\tau$ is a measure of the association between the rank order of the observed value and its position in time. As such, negative values indicate a decreasing trend prior to local bifurcation, while positive values indicate an increasing trend prior to local bifurcation. When observed values are tied, Kendall's $\tau_b$ is calculated, as described by Kendall, 1945 [65].

**Assessing significance of empirical detections of early warning signals.** After analysing the empirical data, we assessed the significance of our detections in four ways. In each case, we used the first half (days one to sixty) of each time series (in warming populations, the pre-bifurcation time series) to generate null distributions of the trend coefficient, against which we compared the observed value of the trend coefficient.

First, we calculated the partial autocorrelation function of the first half of each empirical time series and identified a first order autoregressive model as an appropriate starting point. We then used the first half of each time series to

parameterize an autoregressive model (AR-1) (Eqn. (2.1)) and simulate one thousand real-valued surrogate time series per experimental population [37,66]. Within the autoregressive model, $\beta_1$ is the lag-1 autocorrelation of the residual time series, $\beta_0$ is equal to $\mu(1 - \beta_1)$ where $\mu$ is the mean of the time series, and $\sigma$ is determined via $v$, the variance of the time series, such that $\sigma^2 = v(1 - \beta_1^2)$. $\varepsilon_t$ is uncorrelated Gaussian noise. We then analysed the surrogate time series as described above and summarized the resulting trend coefficients as null distributions. We calculated the probability of detecting the observed trend coefficients by chance as the proportion of surrogate trend coefficients equal to or more extreme than the observed value [37]. In the interest of robustness, we also used a second order autoregressive model to simulate real-valued surrogate time series, and found that resulting null distributions and $P$-values led to the same conclusions. While using an autoregressive model to generate surrogate time series produces real-valued data (rather than non-negative, integer-valued), we opted to include these data to maintain the autocorrelation of the empirical time series, and because three of the four significance assessments we describe and apply produce non-negative integers, offering important points of comparison.

$$x_{t+1} = \beta_1 \chi_t + \beta_0 + \sigma \varepsilon_t \tag{2.1}$$

Second, we used the first half of each time series to parameterize an autoregressive model (AR-1), which we combined with a Poisson process to simulate one thousand non-negative, integer-valued surrogate time series per population. We used 'rstan' [67] to estimate $\beta_1$, $\mu_t$ (to account for non-stationarity), and $\sigma$ such that $\mu_t \sim Normal(\beta_1 \cdot \mu_{t-1}, \sigma)$, and the number of infected individuals was Poisson distributed such that $y_t \sim Poisson(e^{\mu_t})$ (S6 Fig). To fit the model to each experimental time series, we ran four Markov chains for four thousand iterations (S6 Fig). In all cases, resulting $\hat{R}$ statistics indicated model convergence (S6 Fig). We then analysed the surrogate time series, summarized the resulting trend coefficients as null distributions, and calculated the probability of detecting the observed trend coefficients by chance as described above. We also used a negative binomial distribution to simulate surrogate time series as described above. We found that both processes produced similar time series, and that resulting $P$-values led to the same conclusions.

Third, we used an alternative method of calculating the trend coefficient described by Hamed and Rao (1998) to generate a null distribution. We generated each null distribution by drawing one thousand samples from a normal distribution with a mean of zero and variance given by Eqn (2.2), where $n$ is the number of observations in the first half of each time series, and $\rho_s(i)$ is the autocorrelation of the ranks of the observed statistic (e.g., mean, variance, index of dispersion, etc.) [63,68]. While the null hypothesis of the traditional Mann-Kendall trend test is that data are independent, randomly ordered, and without serial correlation, the null hypothesis of Hamed and Rao's (1998) modified Mann-Kendall trend test is that data do not follow a trend, but are autocorrelated [63]. Given that most empirical data are autocorrelated, the modified Mann-Kendall trend test improves on its predecessor by decreasing the incidence of false positives. We calculated the probability of detecting the observed trend coefficients by chance as described above.

$$Var = \frac{2(2n + 5)}{9n(n-1)} \left(1 + \frac{2}{n(n-1)(n-2)} \sum_{i=1}^{n-1} (n-i)(n-i-1)(n-i-2)\rho_s(i)\right) \tag{2.2}$$

Fourth, we applied a bootstrapping method to the first half of each time series to generate one thousand re-sampled time series per experimental population [37,45]. We analysed each group of bootstrapped time series as described above, and summarized the resulting trend coefficients as null distributions. We calculated the probability of detecting the observed trend coefficients by chance as the proportion of bootstrapped coefficients equal to or more extreme than the observed value [37,45]. Within a population, a trend in an indicator was deemed strong if the observed trend coefficient was equal to or more extreme than 95% of null trend coefficients, across all four testing methods.

Finally, we compared control (constant temperature/non-epidemic) and warming (warming treatment/epidemic emergence) coefficients across simulations and experimental populations by calculating the area under the curve (AUC) statistic. The AUC statistic measures the overlap between null and warming trend coefficient distributions and can be interpreted as the probability that a given warming coefficient will be higher than a given null coefficient [58]. As such, values less than 0.5 suggest that a decrease in the statistical metric indicates emergence, while values greater than 0.5 suggest that an increase in the statistical metric indicates emergence, with more extreme values indicating stronger trends.

**Sampling from simulated data.** After analysing the simulated data, we conducted a secondary analysis to replicate and account for the effects of experimental sampling. To do so, we (1) removed non-sampling days from each simulated time series, leaving us with prevalence counts for every third day, (2) implemented sampling via a binomial process, such that on every remaining day, twelve samples were taken from each population, with probability of infection given by the number of infecteds divided by total population size (N = 170), and (3) re-ran analyses as described above.

## Results

### Warming experiment

The empirical data revealed that during the approach to criticality (days one to sixty), one control population, and between one and two warming populations exhibited increases in mean prevalence and variance (Table 3, Figs 3 and 4). One control population (four) and one warming population (seven) exhibited increases in mean prevalence ($0.000 \leq p \leq 0.055$), while one control population (four) and two warming populations (seven and eight) exhibited increases in variance ($0.002 \leq p \leq 0.050$) (Table 4, Fig 5). With reference to the analysis of mean prevalence, population four exhibited a trend coefficient more extreme than 95% of null trend coefficients across three of four tests, and more extreme than 94.5% of null trend coefficients produced by the remaining test (P-value = 0.055). As such, although increasing mean prevalence and variance appeared to precede epidemic emergence within warming populations, the detection of these trends was inconsistent across warming populations and constituted a false positive in a subset of control populations.

We did not detect any trends in the coefficient of variation, the index of dispersion, first difference variance, or autocovariance, in any experimental population. Given that empirical time series contained repeated zeros, we were not able to calculate coefficients or AUC statistics for skewness, kurtosis, autocorrelation, and decay time.

**Table 3. Trend coefficients and AUC statistics across four control populations ("Pop'n 1" to "Pop'n 4") and four warming populations ("Pop'n 5" to "Pop'n 8"), as calculated from empirical time series during the approach to criticality. Each statistical metric was calculated within fifteen-day sliding windows, throughout the pre-critical interval. Given that the temperature of the system increased to 12°C on day sixty, we also considered three pre-critical intervals: Days 1 to 60, Days 20 to 60, and Days 30 to 60. Here, we show analyses of empirical time series performed within the sixty-day pre-critical interval. See analyses within the forty- and thirty-day pre-critical intervals in S7 Fig. To evaluate trends in these metrics, we calculated Kendall's rank correlation coefficient during the pre-critical interval. Negative values indicate a decreasing trend prior to local bifurcation, while positive values indicate an increasing trend prior to local bifurcation. We compared control (constant temperature, non-epidemic) and warming (warming treatment, epidemic emergence) coefficients across simulations and experimental populations by calculating the area under the curve (AUC) statistic. Values less than 0.5 suggest that a decrease in the statistical metric indicates emergence, while values greater than 0.5 suggest that an increase in the statistical metric indicates emergence, with more extreme values indicating stronger trends. Mean, variance, and the index of dispersion exhibit particularly high AUC statistics, and were thus subjected to additional analysis as described in the subsection "Assessing Significance of Empirical Detections of Early Warning Signals".**

| | Pop'n 1 | Pop'n 2 | Pop'n 3 | Pop'n 4 | Pop'n 5 | Pop'n 6 | Pop'n 7 | Pop'n 8 | AUC |
|---|---|---|---|---|---|---|---|---|---|
| Mean | 0.053 | -0.095 | 0.737 | 0.768 | 0.276 | 0.714 | 0.829 | 0.647 | 0.625 |
| Variance | 0.053 | -0.095 | 0.730 | 0.768 | 0.276 | 0.726 | 0.794 | 0.761 | 0.688 |
| Coefficient of Variation | 0.053 | 0.000 | 0.556 | 0.322 | 0.276 | 0.494 | 0.349 | -0.084 | 0.500 |
| Index of Dispersion | 0.053 | 0.000 | 0.556 | 0.322 | 0.276 | 0.648 | 0.349 | 0.274 | 0.688 |
| First Difference Variance | -0.063 | -0.089 | 0.244 | 0.019 | 0.078 | 0.284 | -0.014 | -0.269 | 0.563 |
| Autocovariance | -0.065 | -0.184 | -0.378 | -0.259 | -0.231 | 0.000 | -0.253 | -0.425 | 0.500 |

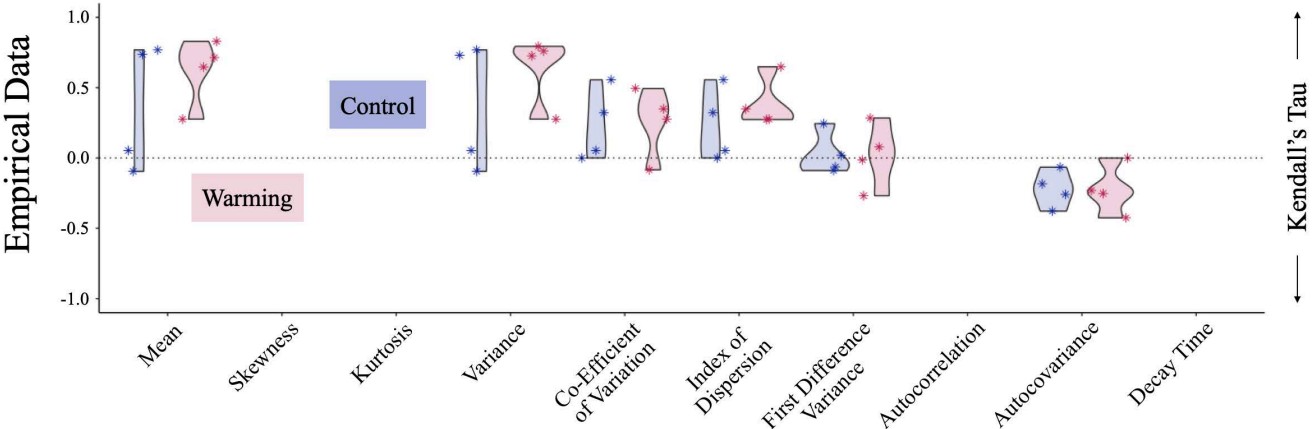

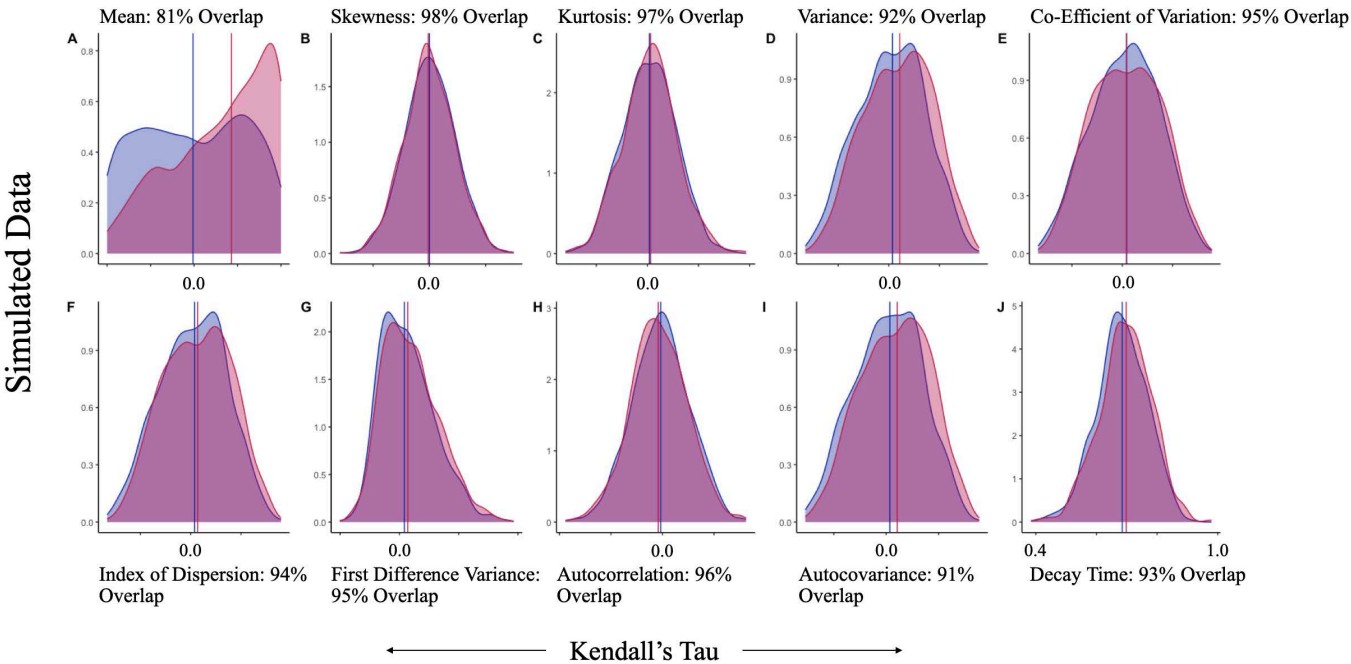

**Fig 3. Violin (top panel) and density (bottom panel) plots showing trend coefficients resulting from analysis of empirical and simulated data, respectively, within a fifteen-day sliding window, between days one and sixty.** In all plots, blue represents "Control" results and pink represents "Warming" results. Violin plots show the trend coefficients resulting from analysis of four populations (1–4) maintained under constant conditions, and four populations (5–8) subjected to a warming treatment. Trend coefficients are shown on the vertical axis and EWS are shown on the horizontal axis. Density plots show trend coefficients resulting from the analysis of one thousand simulations under constant conditions, and one thousand simulations under warming conditions (window size = fifteen days). Density is shown on the vertical axis and trend coefficients are shown on the horizontal axis. Vertical lines represent the median trend coefficient.

### Simulation experiment

**Complete simulated time series.** Simulated data revealed that as warming populations approached critical transition ($R_0 = 1$), mean prevalence and variance increased (Table 5, Figs 3 and 4). Across window sizes and pre-critical intervals, analyses of mean prevalence and variance revealed AUC statistics between 0.651 and 0.686, and 0.558 and 0.632,

**Fig 4. AUC statistics comparing statistical trends in control and test populations.** To evaluate statistical trends, we calculated Kendall's rank correlation coefficient during the pre-critical interval (here, days one to sixty), and compared control (constant temperature, non-epidemic) and warming (warming treatment, epidemic emergence) coefficients across simulations and experimental populations by calculating the area under the curve (AUC) statistic. Values less than 0.5 suggest that a decrease in the statistical metric indicates emergence, while values greater than 0.5 suggest that an increase in the statistical metric indicates emergence, with more extreme values indicating stronger trends. AUC statistics are shown on the vertical axis. EWS are shown on the horizontal axis. Sliding window sizes used to calculate statistical metrics are denoted in the bottom-right of each plot.

respectively (Table 5, S1 Table). We also identified slight increases in the index of dispersion (AUC = 0.518–0.553) and autocovariance (AUC = 0.518–0.562). In all cases, control and warming populations exhibited largely overlapping trend coefficient distributions (mean prevalence: 79% to 81%; variance: 91% to 92%) (Fig 3, S8 Fig). Different window sizes and pre-critical intervals produced slightly different effect sizes, and identified directionally opposing trends in four indicators: kurtosis, skewness, the coefficient of variation, decay time (S1 Table, S8 Fig).

**Subsampled simulated time series.** After accounting for the effects of experimental sampling, warming populations continued to exhibit an increase in mean prevalence during the first half of the experiment (days one to sixty) (AUC = 0.622). In contrast to our analysis of the Complete Simulated Time Series, we identified a slight decrease in variance (AUC = 0.463), a slight increase in kurtosis (AUC = 0.560), and slight decreases in the coefficient of variation (AUC = 0.428), the index of dispersion (AUC = 0.440), and autocorrelation (AUC = 0.446) (S2 Table, S9 Fig).

## Discussion

We analysed experimental and simulated time series of disease spread within *Daphnia magna–Ordospora colligata* populations for EWS of epidemic emergence. While control populations were maintained at a temperature that prevented epidemic emergence, warming populations were subjected to a warming treatment that forced the system through a critical transition to an epidemic state. We found that EWS of epidemic emergence were nearly as likely to be detected in control populations as they were to be detected in warming populations. Our work constitutes the first experimental study of EWS of epidemic emergence [69–71] and provides theoretical and empirical evidence that EWS may not be a reliable indicator of climate-mediated epidemic emergence.

Our analysis of the empirical data revealed that during the first half of the time series, one control population and one warming population exhibited increases in mean prevalence, and one control population and two warming populations exhibited increases in variance (Tables 3 and 4, Figs 3 and 4). Our analysis of the simulated data revealed that as warming populations approached critical transition, they exhibited increases in mean prevalence and variance (Table 5, Figs 3 and 4). We also identified smaller increases in the index of dispersion and autocovariance. However, control and warming populations exhibited largely overlapping trend coefficient distributions (Fig 3, S8 Fig), suggesting that sub- and pre-epidemic trends may not be easily differentiable. In general, agreement between the empirical and simulated data should

Table 4. Detectability of EWS of epidemic emergence in experimental populations[a]. We report the observed trend coefficient, the proportion of null trend coefficients that are more extreme than the observed trend coefficient ("Probability"), and the 95th percentile.

| Indicator | Pop'n | Observed Value | Surrogate Time Series (AR-1 Model) | | Surrogate Time Series (AR-1/Poisson Model) | | Modified Mann-Kendall Test | | Re-Sampled Time Series | |
|---|---|---|---|---|---|---|---|---|---|---|
| | | | Prob. | 95th % | Prob. | 95th % | Prob. | 95th % | Prob. | 95th % |
| Mean | 1 | 0.053 | 0.453 | 0.619 | 0.361 | 0.669 | 0.432 | 0.536 | 0.436 | 0.695 |
| | 2 | -0.095 | 0.604 | 0.619 | 0.390 | 0.578 | 0.625 | 0.569 | 0.576 | 0.676 |
| | 3 | 0.737 | **0.011** | 0.619 | 0.115 | 0.782 | **0.036** | 0.655 | **0.022** | 0.669 |
| | 4 | **0.768** | **0.006** | **0.619** | **0.055** | **0.772** | **0.025** | **0.630** | **0.022** | **0.690** |
| | 5 | 0.276 | 0.280 | 0.600 | 0.595 | 0.647 | 0.214 | 0.586 | 0.375 | 0.690 |
| | 6 | 0.714 | **0.028** | 0.638 | 0.178 | 0.822 | **0.042** | 0.655 | **0.034** | 0.675 |
| | 7 | **0.829** | **0.000** | **0.601** | **0.032** | **0.798** | **0.024** | **0.673** | **0.008** | **0.692** |
| | 8 | 0.647 | **0.037** | 0.619 | 0.102 | 0.753 | **0.051** | 0.648 | 0.060 | 0.670 |
| Variance | 1 | 0.053 | 0.455 | 0.581 | 0.358 | 0.669 | 0.432 | 0.536 | 0.436 | 0.704 |
| | 2 | -0.095 | 0.597 | 0.581 | 0.395 | 0.586 | 0.625 | 0.569 | 0.574 | 0.673 |
| | 3 | 0.730 | **0.015** | 0.620 | 0.095 | 0.768 | **0.038** | 0.656 | **0.020** | 0.653 |
| | 4 | **0.768** | **0.006** | **0.562** | **0.050** | **0.768** | **0.025** | **0.630** | **0.021** | **0.683** |
| | 5 | 0.276 | 0.258 | 0.581 | 0.605 | 0.647 | 0.214 | 0.586 | 0.376 | 0.690 |
| | 6 | 0.726 | **0.019** | 0.619 | 0.148 | 0.816 | **0.039** | 0.656 | **0.038** | 0.687 |
| | 7 | **0.794** | **0.002** | **0.562** | **0.046** | **0.793** | **0.030** | **0.675** | **0.009** | **0.676** |
| | 8 | **0.761** | **0.007** | **0.581** | **0.043** | **0.747** | **0.034** | **0.667** | **0.019** | **0.675** |
| Index of Dispersion | 1 | 0.053 | 0.429 | 0.582 | 0.373 | 0.647 | 0.432 | 0.536 | 0.428 | 0.647 |
| | 2 | 0.000 | 0.494 | 0.600 | 0.336 | 0.498 | 0.505 | 0.513 | 0.501 | 0.600 |
| | 3 | 0.556 | 0.071 | 0.619 | 0.209 | 0.703 | 0.081 | 0.646 | 0.064 | 0.586 |
| | 4 | 0.322 | 0.203 | 0.600 | 0.344 | 0.689 | 0.181 | 0.606 | 0.211 | 0.591 |
| | 5 | 0.276 | 0.174 | 0.467 | 0.600 | 0.628 | 0.214 | 0.586 | 0.362 | 0.649 |
| | 6 | 0.648 | **0.025** | 0.562 | 0.104 | 0.717 | **0.044** | 0.624 | **0.033** | 0.597 |
| | 7 | 0.349 | 0.194 | 0.600 | 0.345 | 0.705 | 0.170 | 0.619 | 0.168 | 0.571 |
| | 8 | 0.274 | 0.242 | 0.581 | 0.295 | 0.617 | 0.187 | 0.522 | 0.244 | 0.611 |

[a]Blue text indicates control populations, while red text indicates warming populations. Yellow cells denote populations in which a given indicator exhibited a trend during the approach to criticality. In all highlighted cases, a trend was detected across four distinct and independent methods.

increase confidence in our ability to use trait-based mechanistic models to study temperature-mediated disease emergence in the *Daphnia magna–Ordospora colligata* system.

Analyses of the empirical data revealed that EWS of epidemic emergence were nearly as likely to be detected in control populations as they were to be detected in warming populations. We suspect that this outcome may be a consequence of the experimental sampling regime and resulting incidence data, which show that during the approach to criticality (days one to sixty), samples from control and warming populations consistently identified comparable numbers of infected individuals. During the approach to criticality, samples from control populations one and two identified between zero and one infected individuals, while samples from control populations three and four identified between zero and two infected individuals. Comparatively, samples from warming populations five and six identified between zero and three infected individuals, while samples from warming populations seven and eight identified between zero and five infected individuals. Additionally, as subsets of experimental populations, experimental samples produced prevalence counts that oscillated between zero and the maximum detected prevalence (Fig 2), indicating that detected prevalence may have

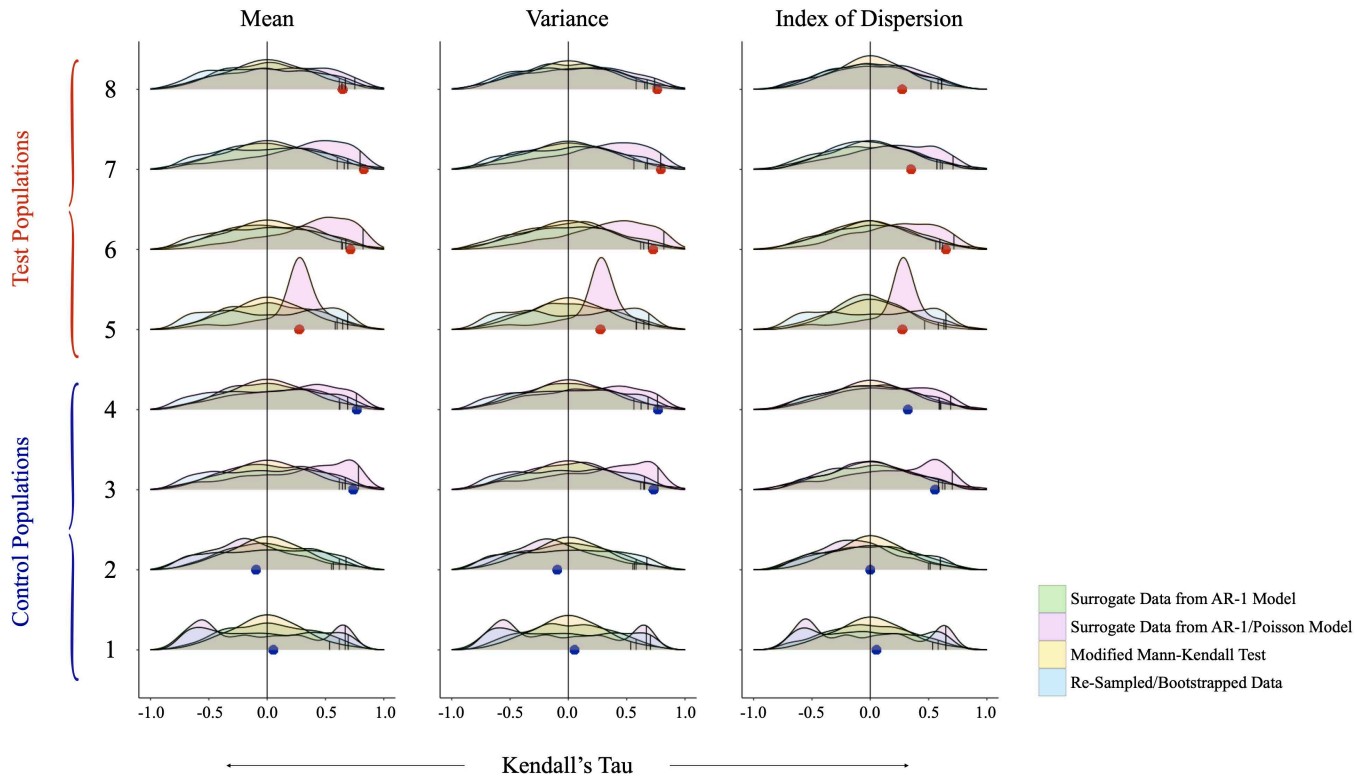

**Fig 5. Null distributions to assess detectability of EWS of epidemic emergence in experimental populations.** "Control" (Populations 1–4) and "Warming" (Populations 5–8) results are differentiated on the vertical axis. Kendall's Tau values are shown on the horizontal axis. Green density plots show one thousand Kendall's Tau values resulting from the simulation of an AR-1 model. Pink density plots show one thousand Kendall's Tau values resulting from the simulation of an AR-1 model with a Poisson process. Yellow density plots show one thousand Kendall's Tau values resulting from the application of a modified Mann-Kendall test. Blue density plots show one thousand Kendall's Tau values resulting from re-sampled time series. Small black vertical ticks represent the 95th quantiles (four ticks for each population, with one tick for each of four tests), while blue ("Control") and red ("Warming") dots show observed values resulting from the empirical data. Labels above each set of density plots denote the EWS being evaluated.

underestimated true prevalence (as observed in Fig 2, which compares experimental detected prevalence and simulated prevalence), and masked statistical trends.

Within the simulation experiment, the inclusion of a sampling process revealed that the granularity of prevalence data can drive the magnitude and direction of observed trends (Fig 4, S9 Fig). While the influence of data quality on the detectability of EWS has been previously discussed [51,53,58,72–74], our work demonstrates that sampling regimes (i.e., how often sampling occurs, how many samples are collected) are particularly consequential. As such, researchers working with imperfect data (experimental or observational) may benefit from performing power analyses to determine the quality of data required to reliably detect early warning signals. While we did not explicitly consider different types of observation error, additional sources of error would increase the difficulty of detecting EWS.

Given that early warning signals are detected by identifying trends in statistical metrics across sliding windows, small changes and oscillations in prevalence counts are consequential, especially when detected prevalence is low. For example, control populations three and four exhibited a modest jump in detected prevalence from zero to one before day sixty. Despite the fact that both populations exhibited a decrease in detected prevalence after day sixty, this jump results in an increase in mean prevalence, and consequently, the detection of mean prevalence as an early warning signal. Similarly, control populations three and four exhibit their first detection after day twenty. Between first detection and day sixty, both

**Table 5. AUC statistics as calculated from simulated time series.** Each statistical metric was calculated within sliding windows, throughout the pre-critical interval. We considered five-, fifteen-, and thirty-day sliding windows. Given that the temperature of the system increased to 12°C on day sixty, we also considered three pre-critical intervals: Days 1 to 60, Days 20 to 60, and Days 30 to 60. To evaluate trends in these metrics, we calculated Kendall's rank correlation coefficient during the pre-critical interval, and compared control (constant temperature, non-epidemic) and warming (warming treatment, epidemic emergence) coefficients across simulations and experimental populations by calculating the area under the curve (AUC) statistic. Values less than 0.5 suggest that a decrease in the statistical metric indicates emergence, while values greater than 0.5 suggest that an increase in the statistical metric indicates emergence, with more extreme values indicating stronger trends.

| Sliding Window Size | 5 Days | | | 15 Days | | | 30 Days |
|---|---|---|---|---|---|---|---|
| Pre-Critical Interval | Days 1–60 | Days 20–60 | Days 30–60 | Days 1–60 | Days 20–60 | Days 30–60 | Days 1–60 |
| Mean | 0.670 | 0.686 | 0.673 | 0.651 | 0.666 | 0.667 | 0.656 |
| Skewness | 0.527 | 0.503 | 0.502 | 0.496 | 0.483 | 0.492 | 0.490 |
| Kurtosis | 0.521 | 0.501 | 0.537 | 0.500 | 0.511 | 0.498 | 0.513 |
| Variance | 0.632 | 0.597 | 0.567 | 0.578 | 0.564 | 0.560 | 0.558 |
| Coefficient of Variation | 0.492 | 0.488 | 0.483 | 0.504 | 0.514 | 0.525 | 0.518 |
| Index of Dispersion | 0.553 | 0.536 | 0.518 | 0.541 | 0.539 | 0.543 | 0.539 |
| First Difference Variance | 0.525 | 0.507 | 0.511 | 0.525 | 0.517 | 0.516 | 0.520 |
| Autocorrelation | 0.509 | 0.516 | 0.503 | 0.506 | 0.513 | 0.523 | 0.500 |
| Autocovariance | 0.537 | 0.531 | 0.518 | 0.562 | 0.549 | 0.549 | 0.549 |
| Decay Time | NA | NA | NA | 0.532 | 0.528 | 0.528 | 0.492 |

populations experience oscillations in detected prevalence between zero and one, resulting in an increase in variance over time. As such, low and oscillating prevalence counts result in the detection of false positives in control populations, calling into question the reliability of early warning signals detected in warming populations. The impact of data quality on the reliability of EWS is also evident in our analysis of the empirical time series within forty- and thirty-day pre-critical intervals. Given the length and structure of the empirical time series, searching for EWS within smaller pre-critical intervals required us to calculate trend coefficients based on very few data points, exacerbating the impacts of process-independent oscillations, and resulting in unreliable and nonsensical trend coefficients and AUC statistics (S7 Fig).

Data quality is critical to the reliable detection of EWS [51,53,58,72–74]. However, epidemiological data are often imperfect. Specifically, they are often structured by aggregate reporting [58] and subject to underreporting, error, and noise [75–79]. The low and oscillating prevalence counts that limit the utility of our experimental data mirror these issues, and as such, confirm the importance of robust (large and frequent samples in experimental studies to avoid repeated zeros, underestimates of prevalence and process-independent oscillations; carefully collected and pre-processed data in observational studies), long-term time series and speak to the circumstances under which these analyses may result in the detection of false positives [34,80]. Most importantly, existing works analyse retrospective time series, and are thus unable to use "control" populations to identify false positives. We identified the same EWS of epidemic emergence in control and warming populations, suggesting that false positives may limit the reliability of EWS as predictors of climate-mediated epidemic emergence.

Our simulation experiment, which mimicked a robust sampling regime, revealed largely overlapping trend coefficient distributions (Fig 3), suggesting that improved sampling may not improve our ability to detect EWS in this system. However, robust and longer-term experimental epidemics in combination with the development of novel methods may still play a role in determining how data quality affects the predictability of epidemic emergence within a range of disease systems [36,81,82]. In addition, while some disease systems, including vector-borne disease systems, seem amenable to this approach [45,47], emerging infectious diseases, like COVID-19 [53,74,83] and systems that exhibit transient dynamics [52] may present difficulties. Further, existing works overwhelmingly detect EWS in disease systems characterised by vector-borne transmission (e.g., dengue, malaria) [45,47,81,84] or direct transmission (e.g., COVID-19, influenza, mumps, pertussis)

[51–53,74,81,83,85]. While increasing concern over the spread of vector-borne diseases in a warming climate [20–22] and the recency of the COVID-19 pandemic may bias researchers towards studying such systems, our findings suggest that predicting the epidemic emergence of environmentally transmitted diseases may present novel difficulties. For example, we may need to consider how relationships between abiotic conditions and pathogen viability in the environmental reservoir affect transmission rates, host population-level dynamics, and subsequently, our ability to detect early warning signals of epidemic emergence [86]. Resolving such difficulties will become increasingly important in the face of increasingly frequent extreme climate events which reduce access to clean water, and promote the spread of water-borne diseases like cholera [87].

In summary, we analysed experimental and simulated time series of disease spread within *Daphnia magna–Ordospora colligata* populations for EWS of epidemic emergence. We identified the same EWS of epidemic emergence in control and warming populations, suggesting that false positives confound the interpretation of EWS in pre-epidemic systems. Given that control populations do not exist in natural systems, our findings suggest that EWS may not be a reliable indicator of climate-mediated epidemic emergence. Our findings also identify potential pitfalls in the detection of EWS in disease systems characterised by environmental transmission, suggesting that the detectability of EWS may vary by transmission mode. As our climate changes, our ability to predict climate-mediated epidemic emergence will become more important than ever. Our findings echo the importance of robust and long-term disease surveillance programs, and demonstrate the need for further investigation via experimental epidemics, and into the predictability of differentially transmitted disease outbreaks.

## Supporting information

**S1 Fig. Empirical and simulated time series pre-processed with bandwidth of two.** Raw data are shown in grey. Pre-processed time series are shown in blue (control populations) and red (warming populations). These data were pre-processed with a Gaussian kernel, with a bandwidth of two. A subset of 200 simulated time series is shown in each of the top panels; experimental time series are shown in the bottom panels.
(PDF)

**S2 Fig. Empirical and simulated time series pre-processed with bandwidth of three.** Raw data are shown in grey. Pre-processed time series are shown in blue (control populations) and red (warming populations). These data were pre-processed with a Gaussian kernel, with a bandwidth of three. A subset of 200 simulated time series is shown in each of the top panels; experimental time series are shown in the bottom panels.
(PDF)

**S3 Fig. Empirical and simulated time series pre-processed with bandwidth of four.** Raw data are shown in grey. Pre-processed time series are shown in blue (control populations) and red (warming populations). These data were pre-processed with a Gaussian kernel, with a bandwidth of four. A subset of 200 simulated time series is shown in each of the top panels; experimental time series are shown in the bottom panels.
(PDF)

**S4 Fig. AUC statistics as calculated from pre-processed empirical and simulated data.** To evaluate statistical trends, we calculated Kendall's rank correlation coefficient during the pre-critical interval (here, Days 1–60), and compared control (constant temperature, non-epidemic) and warming (warming treatment, epidemic emergence) coefficients across simulations and experimental populations by calculating the area under the curve (AUC) statistic. Values less than 0.5 suggest that a decrease in the statistical metric indicates emergence, while values greater than 0.5 suggest that an increase in the statistical metric indicates emergence, with more extreme values indicating stronger trends. AUC statistics are shown on the vertical axis. EWS are shown on the horizontal axis. Sliding window sizes used to calculate statistical metrics are denoted in plot titles. Gaussian kernel bandwidths used to pre-process data are denoted in the bottom-right.
(PDF)

**S5 Fig. Applying six- and thirty-day sliding windows to empirical data.** To evaluate statistical trends, we calculated Kendall's rank correlation coefficient during the pre-critical interval (here, Days 1–60), and compared control (constant temperature, non-epidemic) and warming (warming treatment, epidemic emergence) coefficients across experimental populations by calculating the area under the curve (AUC) statistic. Values less than 0.5 suggest that a decrease in the statistical metric indicates emergence, while values greater than 0.5 suggest that an increase in the statistical metric indicates emergence, with more extreme values indicating stronger trends. AUC statistics are shown on the vertical axis. EWS are shown on the horizontal axis. Sliding window sizes used to calculate statistical metrics are denoted in the legend.
(PDF)

**S6 Fig. Posterior visualizations of autoregressive model with Poisson process.** In (A), (B), and (C), control populations are shown in blue, and warming populations are shown in pink. (A) Per the autoregressive model described in the main text (Eqn. (2.1)), trace plots showing sampled values of $\beta_1$ (lag-1 autocorrelation of the residual time series) and $\sigma$ (variance of the time series) across four chains, over time. (B) Posterior distributions of $\mu$ (mean of the time series). (C) Experimental times series overlaid atop model estimated time series that were used to generate a null distribution of trend coefficients. In (D), we show resulting null distributions of trend coefficients.
(PDF)

**S7 Fig. Trend coefficients resulting from analysis of experimental time series within different pre-critical intervals.** Trend coefficients resulting from analysis of four experimental populations (1–4) maintained under constant conditions (blue), and four experimental populations (5–8) subjected to a warming treatment (pink). Statistical metrics were calculated within fifteen-day sliding windows. To evaluate statistical trends, we calculated Kendall's rank correlation coefficient during the pre-critical interval (forty days (top panel), thirty days (bottom panel), and compared control (constant temperature, non-epidemic) and warming (warming treatment, epidemic emergence) coefficients across simulations and experimental populations by calculating the area under the curve (AUC) statistic. Values less than 0.5 indicate that a decrease in the indicator indicates emergence, while values greater than 0.5 indicate an increasing trend, with more extreme values indicating stronger trends. AUC statistics are shown on the vertical axis. EWS are shown on the horizontal axis.
(PDF)

**S8 Fig. Density plots showing trend coefficient distributions within different pre-critical intervals.** Density plots show the results of one thousand simulations under constant conditions (blue), and one thousand simulations under warming conditions (pink). Density is shown on the vertical axis and trend coefficients are shown on the horizontal axis. Vertical lines represent medians. Statistical metrics were calculated within fifteen-day sliding windows, during sixty-, forty-, and thirty-day pre-critical intervals. The first panel replicates the density plots shown in Fig 3, for reference. While different pre-critical intervals produced comparable overlaps between control and warming trend coefficients, we observed that smaller pre-critical intervals produced larger differences between median control and warming trend coefficients. During the first few days of the experiment, prevalence in control and warming populations was driven by immigration, rather than transmission. Later, and as the system warmed, we expected prevalence in warming populations to be driven by transmission, creating a disparity in prevalence between control and warming populations. As such, smaller pre-critical intervals, which excluded the first fifteen to thirty days of the experiment, produced larger differences between median control and warming trend coefficients.
(PDF)

**S9 Fig. AUC statistics as calculated from simulated data after accounting for the effects of sampling.** To evaluate statistical trends, we calculated Kendall's rank correlation coefficient during the pre-critical interval, and compared control (constant temperature/non-epidemic) and warming (warming treatment/epidemic emergence) coefficients across

simulations and experimental populations by calculating the area under the curve (AUC) statistic. Values less than 0.5 suggest that a decrease in the statistical metric indicates emergence, while values greater than 0.5 suggest that an increase in the statistical metric indicates emergence, with more extreme values indicating stronger trends. AUC statistics are shown on the vertical axis. EWS are shown on the horizontal axis. Analyses were performed within fifteen-day sliding windows, between days one and sixty.
(PDF)

**S1 Table. Median trend coefficients and AUC statistics as calculated from simulated time series within different sliding windows and pre-critical intervals.** Blue text denotes control populations, while red text denotes warming populations. While median tau values are also available in the main text (Table 5), here, we additionally report AUC statistics to directly compare control and warming populations. Each statistical metric was calculated within five-, fifteen-, and thirty-day sliding windows, during sixty-, forty-, and thirty-day pre-critical intervals. To evaluate trends in these metrics, we calculated a median trend coefficient during the pre-critical interval over one thousand control and one thousand warming time series. Negative values indicate a decreasing trend prior to local bifurcation, while positive values indicate an increasing trend prior to local bifurcation. We compared control (constant temperature/non-epidemic) and warming (warming treatment/epidemic emergence) coefficients across simulations and experimental populations by calculating the area under the curve (AUC) statistic. Values less than 0.5 suggest that a decrease in the statistical metric indicates emergence, while values greater than 0.5 suggest that an increase in the statistical metric indicates emergence, with more extreme values indicating stronger trends.
(PDF)

**S2 Table. Median trend coefficients and AUC statistics as calculated from simulated time series after accounting for effects of experimental sampling during the sixty-day pre-critical interval.** Blue text denotes control populations, while red text denotes warming populations. To evaluate trends in these metrics, we calculated a median trend coefficient during the pre-critical interval over one thousand control and one thousand warming time series. Negative values indicate a decreasing trend prior to local bifurcation, while positive values indicate an increasing trend prior to local bifurcation. We compared control (constant temperature, non-epidemic) and warming (warming treatment, epidemic emergence) coefficients across simulations and experimental populations by calculating the area under the curve (AUC) statistic. Values less than 0.5 suggest that a decrease in the statistical metric indicates emergence, while values greater than 0.5 suggest that an increase in the statistical metric indicates emergence, with more extreme values indicating stronger trends.
(PDF)

**S1 Text. Description of data pre-processing.**
(PDF)

## Author contributions

**Conceptualization:** Madeline Jarvis Cross, Devin Kirk, Péter K. Molnár, Martin Krkošek.

**Data curation:** Madeline Jarvis Cross, Devin Kirk.

**Formal analysis:** Madeline Jarvis Cross, Devin Kirk.

**Funding acquisition:** Martin Krkošek.

**Investigation:** Madeline Jarvis Cross, Devin Kirk, Leila Krichel, Pepijn Luijckx.

**Methodology:** Madeline Jarvis Cross, Devin Kirk, Leila Krichel, Pepijn Luijckx.

**Project administration:** Madeline Jarvis Cross.

**Supervision:** Martin Krkošek.

**Visualization:** Madeline Jarvis Cross.

**Writing – original draft:** Madeline Jarvis Cross.

**Writing – review & editing:** Madeline Jarvis Cross, Devin Kirk, Leila Krichel, Pepijn Luijckx, Péter K. Molnár, Martin Krkošek.

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
