## [Decision Letter · Decision Letter 0]

16 Jun 2025

PGPH-D-25-01180

Early warning signals do not predict a warming-induced experimental epidemic

Dear Dr. Jarvis Cross,

Thank you for submitting your manuscript to PLOS Global Public Health. After careful consideration, we feel that it has merit but does not fully meet PLOS Global Public Health’s publication criteria as it currently stands. Therefore, we invite you to submit a revised version of the manuscript that addresses the points raised during the review process.

I would suggest the authors to revise the manuscript by clarifying several aspects including rationale of the study design, model assumption, initial conditions and result interpretation for further consideration.

We look forward to receiving your revised manuscript.

Kind regards,

Sheikh Taslim Ali, M.Sc., Ph.D.

Academic Editor

Journal Requirements:

1. Please note that PLOS Global Public Health has specific guidelines on code sharing for submissions in which author-generated code underpins the findings in the manuscript. In these cases, all author-generated code must be made available without restrictions upon publication of the work. Please review our guidelines at https://journals.plos.org/globalpublichealth/s/materials-and-software-sharing#loc-sharing-code and ensure that your code is shared in a way that follows best practice and facilitates reproducibility and reuse.

Additional Editor Comments (if provided):

Reviewers' comments:

Reviewer's Responses to Questions

**Comments to the Author**

1. Does this manuscript meet PLOS Global Public Health’s publication criteria?

Reviewer #1: Yes

Reviewer #2: Yes

2. Has the statistical analysis been performed appropriately and rigorously?

Reviewer #1: Yes

Reviewer #2: Yes

3. Have the authors made all data underlying the findings in their manuscript fully available (please refer to the Data Availability Statement at the start of the manuscript PDF file)?

Reviewer #1: Yes

Reviewer #2: Yes

4. Is the manuscript presented in an intelligible fashion and written in standard English?

Reviewer #1: Yes

Reviewer #2: Yes

Reviewer #1: The authors analyze experimental and simulated time series of disease spread within populations of the model disease system Daphnia magna–Ordospora colligata for early warning signals of climate-mediated epidemic emergence. They found that the detection of false positives may limit the reliability and utility of early warning signals as predictors of climate-mediated epidemic emergence. The topic is interesting. I have some comments.

Methods

1. Line 130: Please replace “Fig. 1–2” with “Figs. 1–2” and review all similar instances.

2. Line 155: Symbols in “temperature-dependent ( )” brackets is not visible.

3. Line 179: Please provide parameter settings for stochastic simulations.

4. Line 184: Is one thousand time series sufficient?

5. Line 210: What is the purpose of using Gaussian kernel? Have you tried other kernels?

6. Line 221: How did you determine the sliding windows?

7. Line 253: Have you tried other distributions, such as negative binomial?

8. Is it reasonable to additionally provide the confidence interval of AUC?

Results

9. Table 3: Authors should clarify the corresponding scenario of these results (e.g. the sliding window and pre-critical interval), in addition, you should explain what “Pop’n1”-“Pop’n8” stand for. According to Table 2, coefficient of variation = sqrt(variance) / mean, Index of dispersion = variance / mean, which did not coincide with the numbers in Table 3?

Discussion

10. I would like to suggest authors to indicate the situations in which EWS is not a reliable indicator of climate-mediated epidemic emergence and the situations in which EWS may be a reliable indicator.

Reviewer #2: The authors attempt to demonstrate the inferior performance of early warning signals (EWS) as indicators of epidemic emergence, using a model disease system Daphnia magna–Ordospora colligata under control and warming conditions. However, I find that the current version of the manuscript requires several clarifications before it can be considered for publication.

Major Comments

L116: The disease experimental model is outside my expertise, but I am curious about the rationale behind initiating the experiment by introducing only 3 randomly selected adult D. magna from infected stock populations. This design appears to introduce considerable stochasticity, potentially leading to unstable EWS estimates (as discussed around L423). Is there a specific reason for not introducing a larger number of infected individuals from the stock population to improve the stability and reliability of EWS estimates? Given that the population remains endemic below 12°C, increasing the initial infected count might not alter the disease state but could reduce stochastic noise.

L179: I wonder whether the stochastic simulations of the mechanistic model are fully consistent with the experimental initial conditions. Specifically, Fig. S1 shows approximately 20 or fewer infected individuals, whereas the experimental setup began with only 3 infected D. magna at maximum. This discrepancy in initial infection numbers between the simulation and experiment could impact model validity and interpretation.

L191: One model assumption is that R0 = 1 at 12°C theoretically. However, in the experimental system, stochasticity may cause deviations and it is possible for R0 reaching 1 at comparable temperatures (e.g., 11.5°C). Would it be more appropriate to fit a compartmental model directly to the data to estimate the bifurcation time empirically for subsequent analyses?

L234: Please clarify that Kendall’s tau should be the measure of the association between the observed value “of metrics” and its position in time. Given that the observed number of infected is heavily tied, are the derived metrics also subject to many ties? It would be important to clarify how ties are handled in the Kendall’s tau calculations and to assess whether different tie-handling methods affect the results.

L243: The empirical data consist of non-negative integers, but surrogate time series generated by autoregressive (AR) models, the alternative method by Hamed and Rao (L262), and bootstrapping (L273) may include negative values. Please clarify whether these surrogate time series were rounded to integers and how negative values were handled. Since these methods assume continuous or real-valued data, violating the integer constraint of empirical data could bias the null distribution.

L243: Furthermore, why was only an AR(1) model considered instead of higher-order autoregressive model? Besides, since temperature gradually increases in the warming population, the time series may be non-stationary. Incorporating drift terms or applying differencing in the AR(1) model might be more appropriate to capture underlying trends.

L299: The statement that “both control and warming populations exhibited increases in mean prevalence and variance” may be misleading. Focusing on the control populations, only 1 out of 4 control populations showed such increases with statistical significance (p-value). This should be clarified to avoid overgeneralization.

L365, L378: Despite the expected trends in Table 2, what might explain the lack of observed trends in most metrics in both simulations and experiments? A discussion of possible methodological reasons would be helpful.

Minor Comments

L223: Regarding the sliding window analysis, it is unclear why there was insufficient data to apply a 30-day sliding window. Does this imply non-overlapping windows? A sliding window with a 3-day shift, aligned with the experimental sampling frequency, could yield approximately 31 estimates over 120 days, or 22 if excluding overlap around day 60. Additionally, have you considered using a 6-day window (a multiple of 3) instead of 5 days?

L241: It would be more appropriate to refer to values as “observed” or “estimated” rather than “true”, especially since “true” is used extensively throughout the manuscript outside simulation contexts where it may be misleading.

L302: Figure 5, I suggest increasing the plot height for clearer visualization, as the distributions overlap extensively.

L309: Figure 3, please specify in the caption which of the 4 proposed methods was used to generate the null distribution shown.

L355: Table 3, please specify the criteria used to determine whether a trend was detected.

L380: Figure S7, despite comparable overlap percentages, there is a divergence in density near 0 and 1 in the shorter period approaching bifurcation, especially in the null distribution for the mean. Could you provide an explanation for this pattern?

L424–429: Do you mean that the comparable number of samples identified in control and warming populations results in the inferior performance of EWS? Please clarify this point and include the explanation immediately after describing the data.

Supplementary Page 1: Please separate figures and tables into distinct sections. Currently, tables are interspersed among figures which disrupts the flow and readability.

**Do you want your identity to be public for this peer review?** For information about this choice, including consent withdrawal, please see our Privacy Policy

Reviewer #1: No

Reviewer #2: No

---

## [Decision Letter · Decision Letter 1]

31 Aug 2025

Early warning signals do not predict a warming-induced experimental epidemic

PGPH-D-25-01180R1

Dear Ms. Jarvis Cross,

We are pleased to inform you that your manuscript 'Early warning signals do not predict a warming-induced experimental epidemic' has been provisionally accepted for publication in PLOS Global Public Health.

Best regards,

Sheikh Taslim Ali, M.Sc., Ph.D.

Academic Editor

Reviewer #1:

Reviewer #2:

Reviewer Comments (if any, and for reference):

Reviewer's Responses to Questions

**Comments to the Author**

Reviewer #1: All comments have been addressed

Reviewer #2: All comments have been addressed

publication criteria?

Reviewer #1: Yes

Reviewer #2: Yes

3. Has the statistical analysis been performed appropriately and rigorously?

Reviewer #1: Yes

Reviewer #2: Yes

4. Have the authors made all data underlying the findings in their manuscript fully available (please refer to the Data Availability Statement at the start of the manuscript PDF file)?

Reviewer #1: Yes

Reviewer #2: Yes

5. Is the manuscript presented in an intelligible fashion and written in standard English?

Reviewer #1: Yes

Reviewer #2: Yes

Reviewer #1: The authors have addressed all of the comments.

Reviewer #2: The authors have satisfactorily addressed my earlier concerns. I have no additional comments.

**Do you want your identity to be public for this peer review?** For information about this choice, including consent withdrawal, please see our Privacy Policy

Reviewer #1: No

Reviewer #2: No
